# Long-Lasting Hydrophilicity of Al_2_O_3_ Surfaces via Femtosecond Laser Microprocessing

**DOI:** 10.3390/mi17010029

**Published:** 2025-12-26

**Authors:** Alessandra Signorile, Liliana Papa, Marida Pontrandolfi, Caterina Gaudiuso, Annalisa Volpe, Antonio Ancona, Francesco Paolo Mezzapesa

**Affiliations:** 1Intercollegiate Department of Physics “M. Merlin”, University of Bari and Polytechnic of Bari, Via G. Amendola 173, 70125 Bari, Italy; l.papa1@phd.poliba.it (L.P.); m.pontrandolfi@phd.poliba.it (M.P.); caterina.gaudiuso@cnr.it (C.G.); annalisa.volpe@poliba.it (A.V.); antonio.ancona@uniba.it (A.A.); 2National Research Council (CNR), Institute for Photonics and Nanotechnologies (IFN), Via G. Amendola 173, 70125 Bari, Italy

**Keywords:** femtosecond laser microprocessing, laser surface texturing, superhydrophilicity, alumina ceramic (Al_2_O_3_), surface wettability

## Abstract

We explore the wettability modulation induced on alumina (Al_2_O_3_) targets by femtosecond laser texturing to demonstrate the stable and durable hydrophilic character of the surface. Specifically, we identify a suitable operational regime to tailor micro-nanostructures onto Al_2_O_3_ plates and accurately assess the ablation threshold in our experimental conditions. A periodic geometry with triangular patterns of various groove depths, ranging from 3.2 ± 0.1 to 17.1 ± 0.1 µm, was optimized for establishing a long-term wetting response. The latter was monitored on daily basis over a time interval exceeding 40 days by collecting the contact angle measurements of samples with and without a post-process thermal annealing, adopted to stabilize the surface wettability soon after the laser treatment. The results show that deeper grooves significantly enhance and maintain the hydrophilic character, particularly in samples without post-process thermal annealing, where superhydrophilicity (θ < 5°) is demonstrated to persist the entire time throughout the test. These findings disclose the potential for an effective fine-tuning of the alumina wettability, thus opening up the possibility of specific applications requiring long-term control of surface–liquid interactions, such as biomedical implants, and orthopedic and dental prostheses.

## 1. Introduction

Aluminum oxide (Al_2_O_3_), commonly known as alumina, is a ceramic material widely used in numerous technological and industrial applications due to its excellent physical and mechanical properties [1]. Typically, alumina exhibits low density and exceptional hardness, good thermal conductivity, high resistivity, and notable dielectric strength [2,3]. Moreover, from a chemical standpoint, alumina is highly resistant to corrosion and oxidation, showing excellent wear resistance and a high degree of biocompatibility [4]. These properties make alumina an ideal candidate for cutting-edge applications in automotive, aerospace, and biomedicine, where the control of surface wettability is particularly challenging [5,6,7].

Alumina surfaces with hydrophobic behavior, characterized by contact angles greater than 90°, have attracted a lot of attention due to their potential applications in microfluidics, lab-on-chip devices and as functional surfaces for the aerospace and automotive industries [8]. Conversely, alumina hydrophilicity, typically associated with contact angles lower than 90°, has been exploited to promote cell adhesion and osseointegration [9,10], or in crucial aspects of biomedical applications such as joint prostheses (e.g., dental or hip implants) [11,12]. In this context, a comprehensive study of long-term hydrophilicity of alumina appears to be still missing in the current scientific literature.

Over the years, several methods have been developed to adjust the wettability response of alumina via surface modifications, including polymer coatings, UV irradiation, and chemical-assisted and plasma treatments [13,14,15]. However, those approaches often present some limitations in terms of process complexity and fabrication times. Laser processing is known to overcome these disadvantages and intrinsically guarantees improved precision, efficiency, and surface quality [16,17]. Notably, laser processing has been proven to be highly effective in producing hydrophilic surfaces. Triantafyllidis et al. [18] experimentally studied the impact of CO_2_-induced melting and the re-solidification of refractory Al_2_O_3_ composite ceramics, observing a decrease in the fluid contact angle from 41.4° to 26.9°, which they attributed to an increase in surface roughness in relation to the laser power density. However, the authors exclusively focused on analyzing the initial modification of the contact angle, without evaluating the long-term stability of the laser-induced hydrophilicity. Jagdheesh et al. [8] incidentally observed the highly hydrophilic properties on Al_2_O_3_ surfaces immediately after laser texturing with picosecond pulses and Kunz et al. [19] used femtosecond laser pulses to modify the wetting behavior on composite alumina, reporting a significant contact angle reduction but no evidence on hydrophilicity evolution over time. Cao et al. [20] and Wang et al. [21] reported on their experimental investigations with femtosecond laser pulses on alumina, demonstrating a threshold fluence of 10.39 J/cm^2^ and 8.32 J/cm^2^, respectively, and a hydrophilic contact angle of 57.9° after 240 h and 56° after 360 h. However, their results regarding time-dependent hydrophilicity were limited to a period of few days, without addressing whether it would finally stabilize or decay over time.

In this work, we address the entire range of wettability, from hydrophobicity to hydrophilicity, induced by femtosecond laser pulses on alumina surfaces, focusing on the evolution and temporal persistence of surface hydrophilicity over a period of six weeks, without the application of additional surface treatments. To the best of our knowledge, these results represent the first demonstration of such a long-term persistence of laser-induced superhydrophilicity on alumina.

## 2. Materials and Methods

The material used in this study was ADS-96R alumina (Al_2_O_3_) from CoorsTek (Golden, CO, United States), a ceramic substrate widely adopted as an industrial standard [22]. The samples had the dimensions 88.9 mm^2^ × 88.9 mm^2^ and a thickness of 0.64 mm. To remove surface impurities, all the samples were cleaned before laser texturing by ultrasonication in acetone (AR, >99.5%), ethanol (AR, 95%), and distilled water for 15 min each. Finally, the samples were air-dried at 70 °C.

To modify the morphology of the samples’ surfaces, we carried out laser direct writing in ambient air by employing a versatile solid-state laser Pharos SP 1.5 femtosecond laser system (Light Conversion, Vilnius, Lithuania), which offered a wide range of adjustable parameters (pulse duration, fluence, repetition rate, etc.) and significant flexibility for experimental optimization. The pulse duration was set to 190 fs, with a central wavelength of 1030 nm, and the laser repetition rate could be varied from single-shot operation up to 1 MHz. The emitted beam was nearly diffraction-limited (M^2^ = 1.3) and linearly polarized. The laser beam was directed onto the sample surface using a galvo scanner (IntelliSCANN 14, SCAN-LAB, Puchheim, Germany) combined with a telecentric f-theta lens of 100 mm focal length, enabling precise control over the scanning process. The beam waist was measured to be 13.2 ± 0.1 μm (1/e^2^ peak intensity) by a CCD camera FireWire BeamPro (Model 2523 by Photon Inc., Dallas, TX, United States). After laser micromachining, all samples underwent a cleaning procedure consisting of a 15 min ultrasonic bath in isopropyl alcohol to remove any debris and nanoparticles produced during the ablation process.

In the following section, we determined the ablation threshold of alumina (Al_2_O_3_) in our experimental conditions and optimized the surface morphology to demonstrate for the first time, to our knowledge, the stable and durable hydrophilicity induced by femtosecond laser micro-nanotexturing.

## 3. Experimental Section

### 3.1. Ablation Threshold of Alumina with Femtosecond Laser Pulses

We have estimated the ablation threshold of alumina in our experimental conditions using the Liu method [23], which is based on measuring the diameters of laser-induced craters formed on the surface of our samples. According to this method, the relationship between the ablation threshold fluence for N consecutive pulses and the crater sizes is expressed by the following equation:
(1)$$D^{2}=2\;*\;\omega_{0}^{2}\;*\;ln\left(\frac{F_{0}}{F_{th}(N)}\right)$$ where F_0_ is the peak fluence, defined as follows:
(2)$$F_{0}=\frac{2\;*\;E_{p}}{\pi \;*\;\omega_{0}^{2}}$$ where ω_0_ is the Gaussian beam radius at 1/e^2^ and E_p_ is the pulse energy. Through the linear fitting of Equation (1), the radius of the Gaussian laser beam and the threshold fluence for multiple pulses, F_th_(N), were evaluated for each number of pulses N and repetition rate in Table 1. Specifically, seven sets of ablation craters were produced at varying pulse energies, each comprising five matrices. Within each matrix, individual rows were irradiated with an increasing number of laser pulses, while the repetition frequency remained constant within the matrix and varied across adjacent matrices according to the values listed in Table 1 (ranging from 1 to 50 kHz). This procedure was systematically repeated for each matrix in all series. The upper limit of 50 kHz was determined by the maximum pulse energy attainable under the experimental conditions. All the parameters used for each set/matrix/row are summarized in Table 1.

Figure 1a shows the semi-logarithmic graph of the squared crater diameter (D^2^) as a function of pulse energy (E_p_) for different pulse numbers (N) at the highest repetition rate (i.e., 50 kHz). The morphological and optical characterization of the craters was carried out using a Nikon Eclipse ME600 (Nikon Corporation, Tokyo, Japan) optical microscope equipped with ImageJ software (version 1.53k).

It was observed that as both the pulse energy and the number of pulses increased, the squared crater diameter increased linearly, indicating that the ablation area also increased with higher energy. By extrapolating the ablation threshold fluence for each pulse number from the intercept of the linear fitting in Figure 1a, we calculated an average ablation threshold of 1.63 ± 0.16 J/cm^2^.

Figure 1b shows the relationship between the multi-pulse ablation threshold fluence F_th_(N) and the number of laser pulses N at the same repetition frequency of 50 kHz. It was observed that for N values less than or equal to 200 pulses, the threshold fluence F_th_(N) remained nearly constant, in accordance with results shown by Andriukaitis et al. [24], who analyzed the threshold fluence in similar conditions to reveal a rapid saturation in the range 100 ÷ 1000 fs pulses, both in a single-pulse and in a MHz/GHz burst regime. Nonetheless, a slight decrease in the multi-pulse ablation threshold fluence was shown in Figure 1b for N > 200 pulses, which can be rather interpreted as the incubation effect [25,26,27].

### 3.2. Role of Femtosecond Laser Microtexturing on Alumina Wettability

We employed an effective strategy of laser processing [28,29] to access various wettability regimes on Al_2_O_3_ targets and establish the long-term stability of their hydrophilic character. Specifically, we modified the surface morphology with femtosecond laser pulses to create an optimal periodic geometry, which was induced by overlapping cross scanning patterns to generate square-shaped structures, separated by a hatch of 150 μm, followed by a 45° square-shaped pattern.

Figure 2a shows the microstructures and surface morphology, acquired using a profilometer (ContourGT InMotion; Bruker, Tucson, AZ, United States ), with the laser parameters optimized at the maximum available pulse energy of 49.6 µJ for a fixed repetition rate of 50 kHz. The beam scanning speed was set at 20 mm/s to inscribe periodic grooves with a depth of 17.1 ± 0.1 µm (i.e., bottom-1 level in Figure 2a) relative to the intact surface, which, in turn, featured a roughness of about 3.4 ± 0.1 µm. In the 3D image, acquired via the profilometer, different regions of the textured surface are highlighted: the pristine substrate and three areas labeled bottom-1, bottom-2, and bottom-3, corresponding to one, two, and four laser passes over the same spot, respectively. This variation in the number of laser passes was intentionally set in accordance with our laser strategy. Consequently, some areas received multiple overlaps of laser exposure, leading to different groove depths and roughness.

As shown in Figure 2b, higher-resolution investigations by Σigma scanning electron microscopy (SEM) revealed a multi-scale morphology, with a hierarchical decoration of sub-micron ripples and nanoparticles, generally related to thermal effects (melting), as was also demonstrated by Fu et al. [30], who compared surface morphology as a function of repetition rate and pulse overlapping rate, confirming that these two parameters are competing factors responsible for the formation of ripple structures.

We investigated various scanning speeds (ranging from 20 to 750 mm/s) at a fixed pulse energy and repetition frequency, aiming to fine-tune the groove depth only. In Table 2, we summarize the outcome: the groove depth became progressively shallower with increasing scanning speed, as expected [29].

Figure 3 shows how the wetting response of our samples in Table 2 would evolve over a time interval of 6 weeks in air atmosphere. Specifically, the wettability dynamics of laser-textured surfaces with increasing groove depth (see labels 1 to 4 in Figure 3) was assessed by static water contact angle measurements using an Optical Contact Angle Goniometer (OCA 25 by Dataphysics, Filderstadt, Germany). To ensure accurate and repeatable measurements, a set of 3 µL water droplets was deposited on several areas of each sample, and the average value of the resulting contact angles was evaluated. All measurements were performed under identical experimental conditions and following the same measurement protocol to allow direct comparison among different textured regions.

Notably, the samples were stored at room temperature between consecutive measurements, and the contact angles were monitored on daily basis, from day 0 to week 6, although only weekly values have been plotted in Figure 3 for clarity.

The comparison between the contact angle dynamics of both the thermally and non-thermally treated samples is shown in Figure 3a and Figure 3b, respectively. The thermal treatment consisted of heating the samples at 120 ± 1 °C for 3 h in ambient air, as is widely employed to promote the wettability stabilization of similar laser-textured structures [28,29,31,32]. This thermal aging procedure, with rather mild annealing conditions, was here purposely adopted to accelerate the surface enrichment with oxides and organic contaminants, and to expedite the settlement of the wetting response immediately after laser texturing (i.e., without waiting for a standard room temperature aging period of tens of days).

Initially, samples 1–3 in Figure 3a exhibited an almost superhydrophobic regime induced by the thermal treatment, with contact angles as high as 148.9 ± 0.1°, in accordance with results previously observed by Lang et al. in similar conditions [33]. However, such a hydrophobic character was not persistent, with contact angles progressively drifting to a saturation value around 90° within 6 weeks of testing. This behavior is consistent with the time-dependent chemistry stabilization expected at the surface of thermally treated samples [28,30]. Similarly, a comparable trend of monotonic reduction in the contact angle was revealed on pristine (i.e., unstructured) samples subjected to thermal annealing, which displayed an initial contact angle of 83.4 ± 0.1°, reaching a plateau point of 46.3 ± 0.1° after few days. Analogously, a plateau point of 46.1 ± 0.1° was measured on the unstructured and non-thermally treated samples.

Remarkably, sample 4 in Figure 3a displayed unexpected hydrophilicity on day 0, rapidly evolving into a steady superhydrophilic regime which lasted over the testing time and always featured an average contact angle below 5°. We ascribed this behavior to the laser-induced morphology at the surface and speculated on the role played by the grooves of certain depth and roughness, which may have accommodated more fluid in the microstructures, thus promoting the hydrophilic character of the sample. A further plausible explanation for the progressive transition toward hydrophilic behavior may involve changes in the surface’s chemical composition, particularly the accumulation of oxygen-containing functional groups. Recent studies, including those by Cao et al. [20] and Wang et al. [21], have conducted detailed chemical analyses, demonstrating that laser-textured surfaces exhibit elevated atomic percentages of hydroxyl (–OH) and carbonyl (C=O) groups. This enrichment in oxygen content is shown to increase surface energy and, consequently, enhance wettability. Additionally, the formation of sub-micron ripples and nanoparticles, as those shown in Figure 2b, is expected to further affect the surface wettability [34], although behind the scope of this manuscript.

Figure 3b shows the wettability dynamics of non-thermally treated samples: a gradual increase in the contact angle from day 0 to the sixth week was plausibly due to natural aging and surface contamination in air atmosphere. Despite such a gradual increase, all samples except for sample 1 maintained a contact angle below 90° throughout the observation period, demonstrating the persistent hydrophilicity of Al_2_O_3_ surfaces via femtosecond laser texturing. Notably, samples 3 and 4 exhibited a long-lasting superhydrophilic regime (<5°), with their specific morphology fostering very low contact angles in time, with values even below the instrument sensitivity. Although the comparison with sample 3 in Figure 3a may appear to some extent controversial, these preliminary results validated the explicit correlation between fs laser strategy and the long-term superhydrophilicity of alumina, opening up possibilities for the more refined design of micro-nanostructures at the surface in order to establish a robust control of Al_2_O_3_ wettability.

## 4. Conclusions

In summary, this study demonstrates that femtosecond laser micro-nanoprocessing can effectively modulate the wettability of alumina surfaces, enabling the long-lasting hydrophilicity of Al_2_O_3_ substrates for over 40 days, without the need for additional chemical or physical treatments.

The average ablation threshold was accurately determined to be 1.63 ± 0.16 J/cm^2^ under our experimental conditions to identify an optimal operational strategy for fs laser to tailor the surface morphology of alumina. By fine-tuning the pattern design and systematically varying the groove depth and roughness, we established a direct correlation between surface morphology and the wetting response of alumina. Notably, non-thermally treated samples maintained a super-or hydrophilic state throughout the entire testing period, except for samples with shallow grooves. The aging period and the fs laser strategy may be refined to settle the hydrophilic endurance of thermally treated samples, though not for samples with deep grooves.

These preliminary outcomes provide a valuable starting point for future investigations and address key gaps in the current scientific literature, demonstrating the long-term stability of hydrophilic behavior on alumina over much longer period than has previously been reported. Such a capability to control and stabilize hydrophilic states on alumina opens new opportunities for biomedical applications. In particular, enhanced and durable hydrophilicity can promote cell adhesion and osseointegration, which are critical for joint prostheses, such as dental or hip implants, where long-term stability and the reproducibility of wetting properties are essential.

## Figures and Tables

**Figure 1 micromachines-17-00029-f001:**
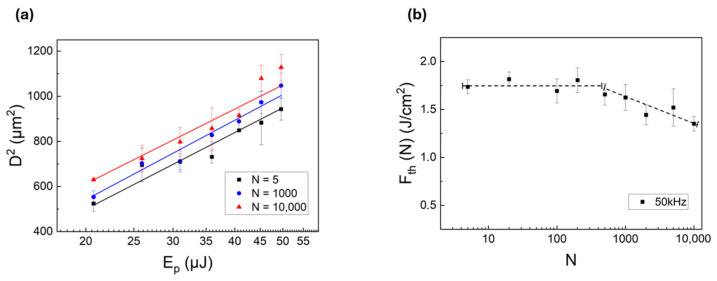
(**a**) Squared crater diameter (D^2^) as a function of the pulse energy (E_p_) for different numbers of pulses (N). The data are displayed on a semi-logarithmic scale, and the curve fitting is performed using the Liu model, which inherently includes the natural logarithm dependence on the fluence. The linear fit shows an R^2^ value higher than 0.97. (**b**) Multi-pulse ablation threshold fluence, F_th_(N), as a function of the number of laser pulses (dashed line is a guide for the eye only). All data refers to a fixed repetition frequency of 50 kHz, which is upper limited by the maximum pulse energy available in our experiments.

**Figure 2 micromachines-17-00029-f002:**
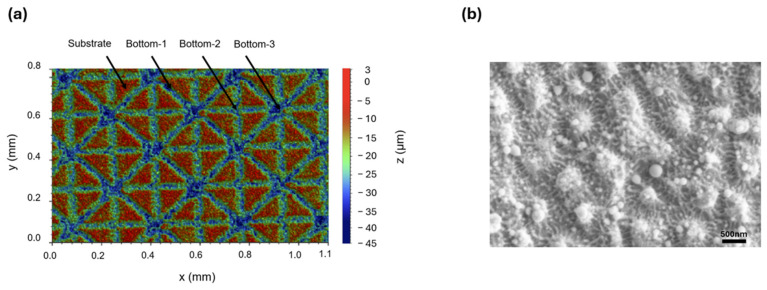
(**a**) Representative 3D topography of laser-inscribed patterns on Al_2_O_3_ surface, featuring a triangular lattice with a texture percentage above 70%. The beam scanning speed was set at 20 mm/s and the depth of the periodic grooves were 17.1 ± 0.1 µm relative to the intact surface. Different regions of the textured surface are highlighted: the unetched substrate (in red), and three areas labeled bottom-1, bottom-2, and bottom-3, corresponding, respectively, to one, two, and four laser passes over the same spot. (**b**) Sub-micron investigation of the hierarchical morphology by Σigma scanning electron microscopy (SEM) from Zeiss (Oberkochen, Germany).

**Figure 3 micromachines-17-00029-f003:**
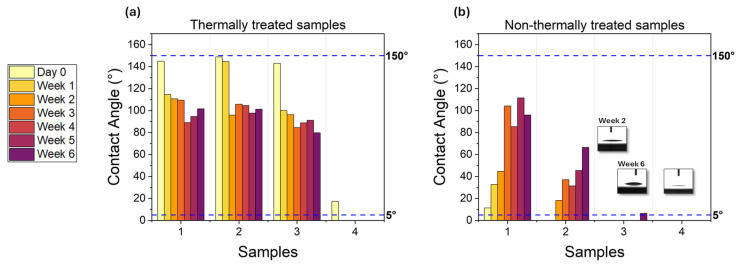
Evolution of the contact angle over time for: (**a**) thermally and (**b**) non-thermally treated samples, with the aging time ranging from day 0 to week 6. Dashed lines indicate the reference threshold angle for superhydrophobic (150°) and superhydrophilic (5°) regimes, respectively. The inset in Figure 3b shows representative images of contact angle measurements (barely above the instrument sensitivity) on the alumina surface at week 2 (i.e., 3.2 ± 0.1°) and week 6 (i.e., 5.6 ± 0.1°), respectively.

**Table 1 micromachines-17-00029-t001:** Experimental parameters used to evaluate the threshold fluence for multiple pulses, F_th_(N), using the Liu method [23].

Parameters	Values
Repetition rates [kHz]	50–25–10–5–1
Number of pulses (N)	5–20–100–200–500–1000–2000–5000–10,000
Pulse energy (E_p_) [µJ]	49.6–45.3–40.8–35.9–31–25.9–20.7

**Table 2 micromachines-17-00029-t002:** Correlation between laser-induced groove depths (corresponding to the measurements at bottom-1 level in Figure 2) and beam scanning speed at a fixed pulse energy of 49.6 µJ, keeping the repetition rate set at 50 kHz.

Sample	Depth [µm]	Scan Speed [mm/s]
1	3.2 ± 0.1	750
2	4.3 ± 0.1	250
3	13.2 ± 0.1	80
4	17.1 ± 0.1	20

## Data Availability

Data will be made available on request.

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
