# Peer review of "Long-Lasting Hydrophilicity of Al2O3 Surfaces via Femtosecond Laser Microprocessing"

_micromachines, 2025, doi:10.3390/mi17010029_

Round 1

Reviewer 1 Report

Comments and Suggestions for Authors

This paper reported the long-lasting hydrophilicity of Al₂O₃ surfaces via femtosecond laser processing. However, some basic knowledge is required to meet the journal.

  1. In introduction part, a systematic literature review is required to conclude why this field is important, what’s the milestones in this field and what is the current challenge. The readers need to know the last 5 years’ literatures about what’s this paper for. However, I didn’t get this point after I checked the reference list from 13-21.
  2. Both laser processing and hydrophilicity have been widely reported. What’s the new in this paper? And how would you declare the results in this paper is long-lasting? What measures are so important? What’s the reason?
  3. To get the threshold, it’s readable on the x-axis (Ln(F0/Fth(N))) by decreasing the D2 to 0. However, I can’t see this in Figure 1. By the way, the x-axis should be Ln(F), according to equation 1. More, why this data is so important to be a Figure? It’s important but it’s the basic knowledge. We need to know the surface morphology, the XRD data, the SEM data and etc.
  4. Some more supporting data is required for Figure 2, as we need to know how is the long-lasting surface. Why the depth is so important? What’s the theory?
  5. Which parameter do you use for the Figure 3? Will the morphology be line or dot at scanning speed 20 mm/s at 50 kHz? This would be tricky and should be discussed, as well as the heat accumulation effects (Chinese Optics Letters 21(5), 051402 (2023)).
  6. How is the focal point? Will it affect the scanning depth and will it break in large area? (Chinese Optics Letters 22(8), 081601 (2024); Nature Photonics, 13, 105–109 (2019))
  7. What’s the difference before and after the thermal-treated measure? Please show the structure morphology, XRD data and other data to support.

Conclusion: The literature is not sufficient to hold their saying and the data is not enough to support their conclusion. However, I would like to review it again if the authors can add the data. End of story.

Reviewer 2 Report

Comments and Suggestions for Authors

This paper investigates the regulation of Al2O3 surface wettability through femtosecond laser microprocessing to achieve long-term hydrophilicity. However, certain sections require supplementation and refinement to enhance the scientific rigor and innovation of the research. The manuscript should be revised and resubmitted for further review. Specific revision suggestions are as follows:

  1. The impact of modifying laser processing parameters on ablation is not intuitive; it is recommended to present morphological and optical characterization results of the craters.
  2. Figure 2 requires additional characterization of surface groove morphology details. The specific morphology of the trench structure can be characterized using electron microscopy or confocal microscopy.
  3. The shapes of each area in Figure 2 are not identical; it is necessary to clarify that contact angle measurements are conducted under a unified standard.
  4. The explanation for why the contact angle changes in the samples is insufficient. It is recommended to characterize the sample surface from the perspective of chemical changes and provide a detailed explanation of the reasons for the altered wettability to enhance the scientific rigor of the paper.
  5. Standardize the drawing, such as modifying the subscripts of parameter names in Figure 1 to match the text.

Reviewer 3 Report

Comments and Suggestions for Authors

Page 2-3 Excellent job reporting section 2 materials and methods.

Page 3 Line 110-112 “Within each matrix, each row was ablated with an increasing number of laser pulses, and the repetition frequency was kept constant and upper limited to 50 kHz…” The beginning of the experimental section is confusing. A figure needs to be provided of the ablation craters and matrices in order to better illustrate the relationship between the various parameters used in each row and matrix. It is currently far too densely presented and is nearly indecipherable upon first or second reading. Particularly, is 50 kHz kept constant or do you use the range of repetition rates you provide in Table 1?  Additionally, if you test more cases besides 50 kHz, does it really matter for this paper? You seem to only then progress with 50 kHz afterwards.

Page 4 Line 146, what is the justification for the texture you produced being optimal? You should either have your own justification or provide something in literature that backs up this claim. Is it optimal because it naturally provides the areas of differing overlap? For wettability, wouldn’t a simple raster scan be optimal?

Page 4 Line 150, again why are the laser parameters being the highest pulse energy for 50 kHz “optimal”. “Optimal” is a large claim to make, what is the justification.

Table 2 – Why did you consider the specific scan speeds that you did, they don’t increase by an obvious constant or multiplicative amount so I’m curious what was the justification. Additionally, which groove depth are you referring to? You just discussed how you have several different areas of varying overlap which lead to different depths. Additionally, this could be another area where you could show a figure comparing your different surfaces on their depth.

Page 4 – Do you clean samples after laser machining? Nanoparticles will have an enormous effect on wettability.

Page 5 line 178 – metrological grade typically refers to the quality of the equipment, not of the measurement. 

Figure 3 and analysis – these results are baseline quite interesting, but there is a significant issue: re-wetting laser samples will alter the wettability response of subsequent tests especially if ablation debris is present that then gets removed with each subsequent wetting test. Chemical interactions between the surface and the test liquid should also be considered. Maintaining the same samples and re-wetting them week after week might alter their long-term performance.

Page 6 Lines 203-207 – neither of your references here indicate time-dependent chemistry stabilization back to 90 degrees. In fact, [30] shows how their samples, after annealing, maintain identical contact angle 14 days later in most cases. Furthermore, both [28] and [30] are works on metals, whereas you are working on a ceramic. I do think your reduced contact angle is likely due to rewetting. If you want to actually create a robust comparison, create six samples of each and test a new one each week. Moreover, if you are not yet cleaning your samples after laser machining, you could consider doing so. One such study that goes into the effects of rewetting (again on metals) is "The Tuning of LIPSS Wettability during Laser Machining and Through Post-Processing."

Reviewer 4 Report

Comments and Suggestions for Authors

The research topic presented in the manuscript is timely, interesting and important. Aluminum oxide is widely used in numerous technological, industrial and medical applications due to its excellent physical, mechanical and biological properties. In these applications, the wettability of the surface being used and its controllable adjustment are often very important. Several previous studies have demonstrated that laser ablation is suitable for creating areas with defined wetting properties on the surface of alumina samples. In the manuscript, the authors investigated the effects of femtosecond laser ablation. The research topic is therefore good, the results achieved are interesting, the figures presented are convincing, and the conclusions drawn from them are valid. However, I believe that the manuscript needs additions. The question raised would be worth a systematic series of investigations or/and interpretation.

Abstract

The abstract could mention the irradiation parameters used during the experiments (wavelength, fluence, frequency, pulse length, etc.).

  1. Introduction

Since there have been quite a few publications on the research topic of the manuscript in recent years, I suggest that the novelty of the manuscript and the reached results should be emphasized more in this chapter.

  1. Experimental section

3.2. Role of femtosecond laser microtexturing on alumina wettability

Did you not perform SEM investigations on the ablated areas? The nanostructure on the formed microstructures can also significantly influence the wetting properties. The manuscript mentions nanostructures in several places, but no studies on this subject are included anywhere. I think it would definitely be worth looking at the ablated areas with a scanning electron microscope.

Figure 2: I can't interpret the color scale. If the red color corresponds to the untreated surface, it should be zero on the z(um) scale. In this case, the etching depths should be calculated from this level.

Table 2: It would have been useful to measure roughness data (Ra) in the treated areas and include it in the table.

Figure 3: I don't understand why the contact angle data for untreated areas were not included in the graphs. This would have proven the effectiveness and justification of the laser treatment.

You have talked about morphology several times, but in fact you have not conducted any morphological investigations. Although I think they would be important. I suggest supplementing the manuscript with profilometer or electron microscopic studies.  

Have you examined whether the ratio of treated/untreated surfaces affects the wetting properties? Does the pattern created by ablation have an influencing effect?

  1. Conclusions

Have you investigated whether morphological changes occur during heat treatment? What do you think causes the significant difference observed between the wettability of heat-treated and untreated surfaces?

What do you think plays the most important role in the wetting enhancement process? The shape, filling, depth or morphology of the ablated structure, or perhaps the change in the chemical composition of the surface layer? Which can be affected by heat treatment?

Have you performed any feasibility biomedical demonstration experiments? I believe that cell adhesion depends on the wettability of the formed surface and the ablation structure.

Reviewer 5 Report

Comments and Suggestions for Authors

The publication presents an informative investigation of the surface hydrophilicity caused by laser ablation. Additionally, data similar to those in Figs. (a and b) should be included in the text by the authors for at least one distinct repetition rate.

Round 2

Reviewer 1 Report

Comments and Suggestions for Authors

I don’t think the paper can be accepted in its current form.

1. The authors replied they have checked through the published papers in the last 5 years and showed 4 as a comparison. This is good. However, they didn’t show why the superhydrophilicity is so important for Al2O3. And what’s the current challenge. The first point is why the authors must do it on Al2O3? The second point is what kinds of micro/nanostructures for superhydrophilicity the authors should do to break through the current bottleneck? To clarify these two points, the readers know why the structures presenting in this paper is long-lasting.

  1. In the second reply, the authors argued a lot and show the CA changing with time. However, the reason is still not discussed, and there is no solid data to support.
  2. Obviously, the D2 is linear with Ln(Ep), or you can display the axis of Ep in Ln, which is the authors’saying semi-logarithmic scale. However, it’s not clear to see it in Fig 1a, which may easily misleading the readers.
  3. There is no new data supplemented for Fig 2.
  4. There is strong debris on the surface of Fig 4. It’s difficult to see the line property and judge if there is heat accumulation effects. Please compare with the provided paper (Chinese Optics Letters 21(5), 051402 (2023)). Don’t argue too much. Heat accumulation can raiseto 3000 degree. And we can’t see your hierarchical surface.
  5. There is no discussion on the two literatures I provided. This is too bad.
  6. It’s so ridiculous to see the authors provides two so large figures to check the morphology. Scientific paper needs enough data.

Overall, the authors argued a lot but provided a little data to support their paper. This is sad. 

Reviewer 2 Report

Comments and Suggestions for Authors

The authors have fully improved the manuscript and answered the question reviewer commented which are acceptable. So I think the manuscript can be accepted.

Author Response

We thank the reviewer for the positive evaluation of our revised manuscript and for recommending acceptance. We appreciate the reviewer’s time and constructive feedback during the review process.

Reviewer 4 Report

Comments and Suggestions for Authors I have checked the authors' responses to the reviewers' questions and comments,
and the revised and supplemented manuscript. In my opinion, these meet the expectations,
and I therefore recommend acceptance of the manuscript.

Author Response

We thank the reviewer for carefully evaluating our responses and revised manuscript, and for recommending acceptance. We are grateful for the valuable comments provided throughout the review process.

Round 3

Reviewer 1 Report

Comments and Suggestions for Authors

The authors have addressed my concerns. However, some key points, such as the differences before and after annealing, were not included. There are no SEM images, and there is not enough discussion on why the results are long-lasting. Without this, it's difficult to inform readers of what has happened. 
